# Contrasting Patterns of Microbial Communities in Glacier Cryoconite of Nepali Himalaya and Greenland, Arctic

**Purnima Singh [1,2], Masaharu Tsuji [3] , Shiv Mohan Singh [4,5,*] and Nozomu Takeuchi [6]**

[1] Parvatibai Chowgule College of Arts & Science, Goa 403602, India; purnimabitsgoa@gmail.com
[2] NPDF-Science and Engineering Research Board, New Delhi 110 070, India
[3] Department of Materials Chemistry, Asahikawa College, National Institute of Technology, Hokkaido 071-8142, Japan; spindletuber@gmail.com
[4] Polar Biology Laboratory, National Centre for Antarctic and Ocean Research, Goa 403804, India
[5] Department of Botany, Institute of Science, Banaras Hindu University, Varanasi 221005, India
[6] Department of Earth Sciences, Graduate School of Science, Chiba University, Chiba 263-8522, Japan; ntakeuch@faculty.chiba-u.jp
* Correspondence: drshivmohansingh@gmail.com

**Abstract:** To understand the microbial composition and diversity patterns, cryoconite granules were collected from two geographical areas, i.e., Nepali Himalaya and Greenland, Arctic. 16S rRNA, ITS and the D1/D2 domain sequencing techniques were used for characterization of microbial communities of the four glaciers. The total 13 species of bacteria such as *Bacillus aryabhattai*, *Bacillus simplex*, *Brevundimonas vesicularis*, *Cryobacterium luteum*, *Cryobacterium psychrotolerans*, *Dermacoccus nishinomiyaensis*, *Glaciihabitans tibetensis*, *Leifsonia kafniensis*, *Paracoccus limosus*, *Polaromonas glacialis*, *Sporosarcina globispora*, *Staphylococcus saprophyticus*, *Variovorax ginsengisoli*, and 4 species of fungi such as *Goffeauzyma gilvescens*, *Mrakia robertii*, Dothideomycetes sp., Helotiales sp. were recorded from Nepali Himalaya. Among these, 12 species of bacteria and 4 species of fungi are new contributions to Himalaya. In contrast to this, six species of bacteria such as *Bacillus cereus*, *Cryobacterium psychrotolerans*, *Dermacoccus nishinomiyaensis*, *Enhydrobacter aerosaccus*, *Glaciihabitans tibetensis*, *Subtercola frigoramans*, and nine species of fungi such as *Goffeauzyma gilvescens*, *Mrakia robertii*, *Naganishia vaughanmartiniae*, *Piskurozyma fildesensis*, *Rhodotorula svalbardensis*, *Alatospora acuminata*, *Articulospora* sp., *Phialophora* sp., *Thelebolus microspores*, and *Dothideomycetes* sp.), were recorded from Qaanaaq, Isunnguata Sermia and Thule glaciers, Greenland. Among these, five species of bacteria and seven species of fungi are new contributions to Greenland cryoconite. Microbial analyses indicate that the Nepali Himalayan cryoconite colonize higher numbers of microbial species compared to the Greenland cryoconite.

**Keywords:** cryoconite; diversity; bacteria; fungi; Greenland; Himalaya

## 1. Introduction

Cryoconite are dark-colored, bio-inorganic dusts, transported by wind and deposited on the glaciers and Sea ice [1]. Once the dust material is settled on glacial surface, it aggregates, becomes spherical-granular structure and absorbs more solar radiation creating melt holes (cryoconite holes) [2,3]. Cryoconite is mainly composed of organic matter such as algae, cyanobacteria, bacteria, fungi, rotifers and their metabolites [4–8], and inorganic matter as a mixture of different elements [4,9–11]. The dark color of cryoconite is due to combinations of minerals, organic matters and microorganisms, which reduces the albedo of glaciers and accelerates ice melting [12–16]. The mineral dust composition, nutrients, climate conditions, and physical features of glaciers altogether affect colonization of the

microbial community in glacier cryoconite holes [13,15,17–21]. Cryoconite can cover about 0.1–10% of the ice surface in the ablation zone of glaciers [6] and is distributed in many parts of the world including Arctic, Antarctic, and Himalaya. Antarctic cryoconite generally remain frozen at the glacier surface while only a few of them are exposed during summer [1,22–24]. Arctic, Alpine and Himalayan cryoconite holes usually remain open during summer seasons [11,25]. Cryosphere constitute a major part of the land and hydrosphere, therefore, research on the cryoconite is of great significance.

Studies on Arctic cryoconite began with the investigation of brown algae [26]. Afterward, cyanobacteria [27,28], viruses and virus-like particles [29,30], diatoms [31], Bacteria [32–34], yeast and filamentous fungi [35–38] were focused on in Svalbard. Cryoconite holes have also been searched from the Greenland Ice Sheet (GrIS) [39,40] and Himalayan glaciers [11].

Greenland Ice Sheet is biologically active, and covers more than 200,000 km$^2$ [41,42]. Several studies were conducted on microbial diversity using non-culturable approach in the area of the GrIS [19,43–51], and reported the presence of bacteria from four different classes such as Proteobacteria, Cyanobacteria, Bacteroidetes and Actinobacteria. Besides, these studies were also conducted on ice algal blooms at GrIS supraglacial habitats [52–56].

The culturable approach has been used for analyses of bacterial diversity from Antarctic cryoconite [57], Svalbard, Arctic [34] and Alpine glacier [58]. Recently, Perini et al. [56] analyzed bacteria and fungi from cryoconite of Greenland Ice Sheet (GrIS), and reported nine genera of culturable bacteria (*Arthrobacter* sp., *Bacillus* sp., *Cryobacterium psychrotolerans*, *Pseudomonas* sp., *Janthinobacterium* sp., *Rugamonas rubra*, *Sphingomonas* sp., *Rhodopseudomonas* sp., *Undibacterium* sp., and 10 genera of culturable fungi (*Aspergillus conicus*, *Aspergillus restrictus*, *Articulospora* sp., *Cladosporium* sp., *Dothiora* sp., *Glaciozyma* sp. nov.1, *Mrakia* sp., *Penicillium bialowiezense*-like, *Preussia* sp., *Rhodotorula svalbardensis*, *Vishniacozyma victoriae*). The studies on culturable microbes of supraglacial ecosystems have immense importance in physiological (growth of the isolates at various pH, salt tolerance ability, optimum temperature for the growth), biochemical (carbon utilization ability, antibiotic resistance patterns, fatty acid profiles) and biotechnological (enzymes, antifreeze proteins etc.) characterizations of individual species [34].

Himalaya is a unique place in the mountain ecosystems of the world. It spreads over 2500 km, starting from west-northwest to the east. Cryoconite holes are also present over the glacier's ablation zone of Himalaya [4,11,12,59]. The different studies were conducted on microbes belonging to Nepali Himalaya such as microbial biomass [60], bacterial diversity [61–63], Algal community, Cyanobacteria and Planktonic diversity [64–66], and microbial ecology of Himalaya [67].

Literature review suggests that a study on culturable microbial communities from cryoconite habitat of Nepali Himalaya is a gap area, and Greenland Arctic are scarce, therefore, investigation is needed. The aim of current study is to investigate microbial communities sustaining in cryoconite holes at two geographical domains such as Greenland and Himalaya. The sustainability of microbial species helps them in colonization, succession and composition of communities in an extreme glacier environment.

## 2. Materials and Methods

### 2.1. Sampling Site

Cryoconite samples used in this study were collected from three Greenland glaciers: Qaanaaq (QG), Thule (TG), Isunnguata Sermia (IS), and one Nepali Himalaya glacier, Yala (YG) (Figure 1a–d). During summer months, a number of cryoconite holes were observed on the ablation zone of all glaciers. Cryoconite were collected (Figure 1e,f) from the bottom of cryoconite holes at each study site (two sites on QG, one site on TG, two sites on IS, and two sites on YG). Cryoconite granules were loose, rounded and brownish-black in color. Cryoconite granules were collected from bottom of cryoconite holes by aspirating with a sterile disposable plastic pipette into sterile tubes, and stored at −20 °C until analyses. Sampling was carried out on 15 August 2008 from Yala glacier, Nepali Himalaya,

on 15 July 2012 from Qaanaaq glacier, and on 15 August 2015 from Thule and Isunnguata Sermia glaciers, Greenland. One to two samples were collected at each site; however, one was used for culture-based study at different dilutions, medias and temperature gradient. The other metadata collected was pH which ranged between 5.8 and 6.2 at Nepali Himalaya and 5.8 to 6.4 at Greenland. Water temperature of all of the cryoconite holes in both Himalaya and Greenland were zero degree. EC of the water ranged 0.6 to 0.8 µS cm$^{-1}$ at Nepali Himalaya and 6.2 to 8.0 µS cm$^{-1}$ at Greenland. NH4 content varied 26.70 ppb at Thule glacier, 1.2–2.4 ppb at Isunnguata Sermia glacier and 0.1 to 28.4 ppb at Qaanaaq glacier. NO3 content of 25.93 ppb at Thule, 12.5–46.8 ppb at Isunnguata Sermia, and 0.0–7.3 ppb at Qaanaaq glacier was recorded.

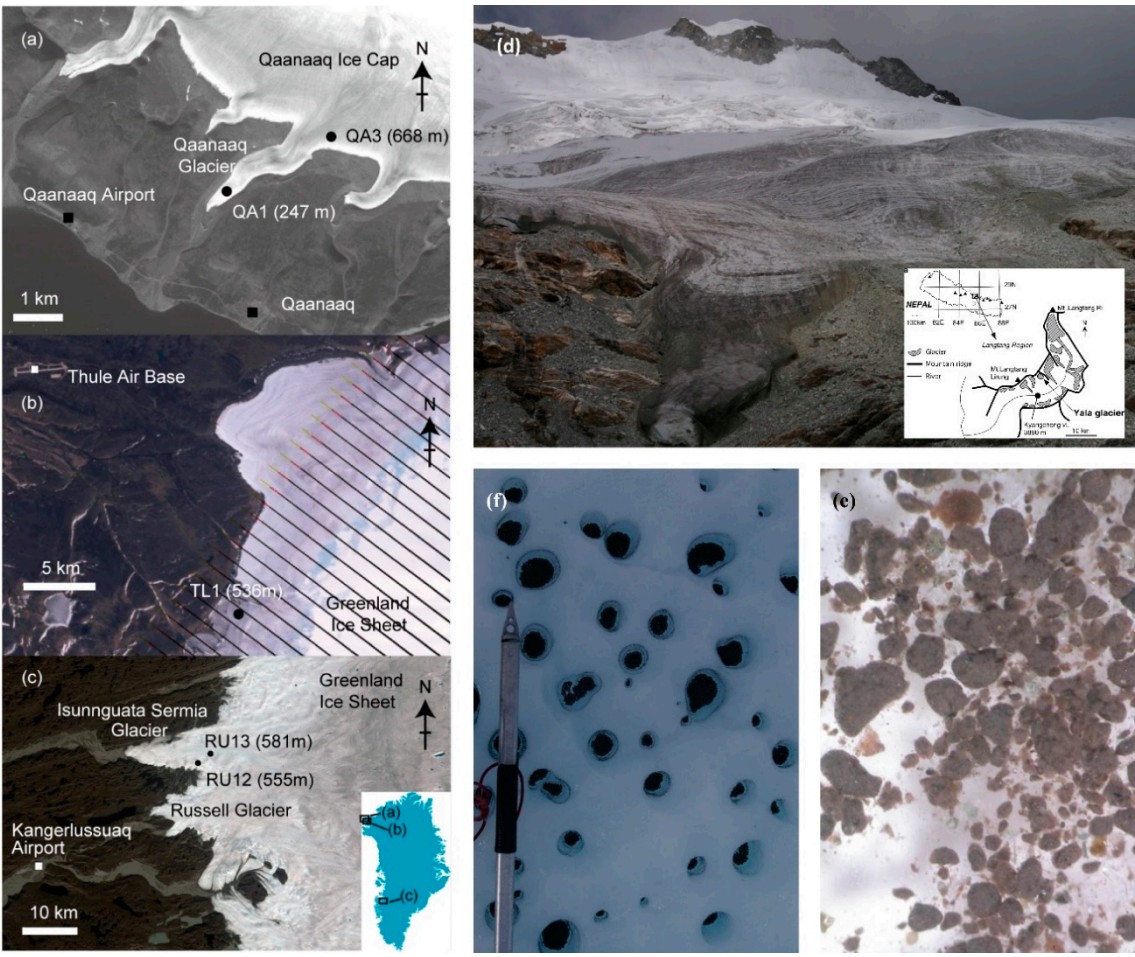

**Figure 1.** Landscape of glaciers and sampling locations: (**a**). Qaanaaq glacier, Greenland,(**b**) Thule glacier, Greenland, (**c**) Isunnguata Sermia glacier, Greenland, (**d**) Yala glacier, Nepali Himalaya, (**e**) Cryoconite holes on Isunnguata Sermia Glacier in Greenland, (**f**) Cryoconite granules collected from Yala Glacier, Nepali Himalaya.

### 2.2. Isolation and Culturing of Bacteria and Fungi

One gram of cryoconite was suspended in 9 mL saline, and serial dilutions procedure ($10^{-1}$, $10^{-2}$ and $10^3$) were followed [34,35]. The spread plate method (0.1 mL) was used on different medias such as Nutrient Agar (NA), 1/10 Nutrient Agar (1/10 NA), Marine Broth (MB), 1/10 Marine Broth (1/10 MB), and R2A Medium. Similarly, 100 µL of cryoconite were plated onto different media such as Rose Bengal Agar (RBA), Potato Dextrose Agar (PDA) and Malt Extract Agar (MEA) by following pour plate and spread plate techniques. Plates were incubated at 1, 4 and 15 °C for 14–30 days. Colony forming units (CFUs) appeared after incubation was observed. Bacterial and fungal isolates showed

unique morphotypes were picked from each plate. Pure cultures were obtained after 2–3 subcultures for further study.

## 2.3. DNA Extraction, Polymerase Chain Reaction (PCR) and Sequencing of Bacteria and Fungi

Representative isolates of bacteria were selected for molecular characterization. The pure cultures were subjected to total DNA extraction. Genomic DNA was isolated following standard protocol [68]. Universal primers 16F27 [5'-CCA GAG TTT GAT CMT GGC TCA G-3'] and 16R1492 [5'-TAC GGY TAC CTT GTT ACG ACT T-3'] were used for PCR amplification of the 16S rRNA gene. The amplified PCR product was purified by PEG-NaCl precipitation and sequenced on an ABI® 3730XL automated DNA sequencer following standard protocol. Assembly was carried out using Lasergene package, and species identification was performed by using EzBioCloud database [69]. Sequences of present study were deposited at National Center for Biotechnology Information (NCBI) GenBank database.

Representative isolates of yeasts and filamentous fungi were also selected for molecular characterization. DNA was extracted using ISOPLANT II (Wako, Japan) following manufacturer's protocol. The extracted DNA was amplified for ITS and D1/D2 regions by PCR, using KOD-plus DNA polymerase (Toyobo, Japan). PCR reaction mixture was consisted of 10–50 ng (extracted DNA, 0.2 mM dNTPs, 2.0 mM $MgSO_4$, 0.3 μM ITS1F (5'-GTAACAAGGTTTCCGT) and NL4 (5'-GGTCCGTGTTTCAAGACGG), and 1 U KOD-plus DNA polymerase). PCR condition (5 min at 94 °C, followed by 30 cycles for 15 s at 94 °C, 30 s at 54 °C and 90 s at 68 °C) were followed. PCR products were checked by electrophoresis using 1.5% (w/v) agarose gel. Amplicons were purified on Sephacryl S-400HR (Sigma-Aldrich, Japan), and sequenced on an ABI Prism 3130xl Sequencer (Applied Biosystems). The sequences of bacteria and fungi were deposited in the DNA data bank (NCBI).

## 2.4. Sequence Alignment and Phylogenetic Analyses of Bacteria and Fungi

Sequence alignments of 16S rRNA gene of different bacterial isolates with the homologous sequences (retrieved from Genbank) were performed using Clustal W Molecular Evolutionary Genetics Analysis (MEGA v4.0.) software [70]. The sequences of the isolates were subjected to a NCBI BLAST search. Sequence similarity for 16S rRNA gene was analyzed and phylogenetic trees were constructed by using Neighbor-joining method [71] and Tamura-Nei model [72]. The bootstrap consensus tree [73] represents the evolutionary history of the taxa analyzed.

ITS and D1/D2 sequences of fungi were aligned using Clustal W and MEGA 7 [74]. The pairwise alignment was performed using EMBOSS Matcher-Pairwise Sequence Alignment tool [74–82]. Phylogenetic trees were constructed by Neighbor-joining method [71] with the Tamura-Nei model. Tree nodes were tested by bootstrap analysis with 1000 replicates, and a bootstrap percentage ≥50%.

## 3. Results

### 3.1. Characteristics of the Microbial Strains

Several bacterial CFUs appearing on the culture media plates varied at each site. It ranged $2.09 \times 10^3$ to $3.26 \times 10^3$ at Nepali Himalaya, while at Greenland, $7.93 \times 10^3$, at Thule glacier, $7.73 \times 10^4$ to $17.02 \times 10^4$ at Isunnguata Sermia glacier, and $3.42 \times 10^4$ to $7.78 \times 10^4$ at Qaanaaq glacier. Cultures showed different colors (red, orange, yellow, white, cream), entire margin, convex, pulvinate, opaque, and smooth texture. Among these, 75 morphologically distinct representative isolates were purified for further analyses. Of these, 36 isolates were from YG, 10 from TG, 12 from IS, and 17 from QG glaciers.

Fungal colonies of the yeast and filamentous fungi were creamish, orange, peach, pink, and white in color, with regular and irregular margin. Yeast cells occurred singly or in groups and their shape were globose or sub-globose. Based on these morphological features, distinct strains were selected for molecular identification. Of these, 15 were from Himalaya, 8 from TG, 22 from IS, and 19 from QG glaciers.

The results of different growth media, and temperature used to obtain the different isolates information included for each isolate are shown in Supplementary Tables S1 and S3. Large number of isolates (139 strains) recorded from Himalaya and Greenland's cryoconite granules were further analyzed.

*3.2. Phylogenetic Analyses of Bacteria*

The identification of bacterial taxa was based on the 16S rRNA gene domain sequencing. EMBOSS Matcher Pairwise Sequence Alignment tool was used for pairwise alignment. Total sequence lengths after alignment, positions with base changes, NCBI sequence deposition numbers, sequence similarities (%), and identification are given in Table 1.

The 16S rRNA gene sequences analyses of different strains (isolates), showed 13 genera belonging to 10 families, namely, Bacillaceae (*Bacillus*), Staphylococcaceae (*Staphylococcus*), Planococcaceae (*Sporosarcina*) affiliated to phylum Firmicutes; Microbacteriaceae *(Cryobacterium, Glaciihabitans, Leifsonia, Subtercola)*; *Dermacoccaceae* (*Dermacoccus*) related to phylum Actinobacteria are Gram-positive, whereas Caulobacteraceae (*Brevundimonas),* Comamonadaceae (*Polaromonas*), Rhodobacteraceae (*Paracoccus*), Rhodospirillaceae (*Enhydrobacter*), Comamonadaceae (*Variovorax*) affiliated to phylum Proteobacteria are Gram-negative. The affiliations of different strains of bacteria with database have been elaborated in phylogenetic trees (Figures 2–5).

The three Gram-positive bacteria (*Bacillus, Staphylococcus* and *Sporosarcina*) belonging to the order *Bacillales* formed separate clades (Figure 2a). The four other Gram-positive bacteria (*Cryobacterium, Glaciihabitans, Leifsonia, Dermacoccus*) were affiliated to phylum *Actinobacteria* of order Micrococcales (Figure 2b,c and Figure 3). One other Gram-positive group *Subtercola*, belonging to the phylum Actinobacteria, formed a separate clade (Figures 4 and 5). The three Gram-negative bacteria (*Brevundimonas, Polaromonas, Variovorax*) belonging to the phylum Proteobacteria formed separate clades (Figure 2d). One other Gram-negative group *Enhydrobacter* was affiliated to the phylum Proteobacteria of class Alphaproteobacteria (Figure 2a,b).

Comparative analyses of four glaciers belonging to Himalaya and Greenland showed contrasting pattern in bacterial species composition. Thirteen species such as *Bacillus aryabhattai, Bacillus simplex, Brevundimonas vesicularis, Cryobacterium luteum, Cryobacterium psychrotolerans, Dermacoccus nishinomiyaensis, Glaciihabitans tibetensis, Leifsonia kafniensis, Paracoccus limosus, Polaromonas glacialis, Sporosarcina globispora, Staphylococcus saprophyticus, Variovorax ginsengisoli* were recorded from Yala glacier, Himalaya (Table 1). In contrast to this, at Greenland, four species such as *Cryobacterium psychrotolerans, Dermacoccus nishinomiyaensis, Enhydrobacter aerosaccus, Glaciihabitans tibetensis* from Thule glacier. Three species such as *Cryobacterium psychrotolerans, Dermacoccus nishinomiyaensis, Subtercola frigoramans* belonging to Isunnguata Sermia glacier, and four species such as *Bacillus cereus, Cryobacterium psychrotolerans, Glaciihabitans tibetensis, Subtercola frigoramans* were recorded from Qaanaaq glacier (Table 1).

*3.3. Phylogenetic Analyses of Fungi*

The results of sequence analyses such as total sequence lengths, positions with base changes, NCBI sequence deposition numbers, sequence similarities (%) and identity of yeasts and filamentous fungi are given in Tables 2 and 3.

**Table 1.** Sampling points, sequence length, NCBI-GenBank accession numbers and species identity of bacterial isolates by 16S rRNA gene sequences similarity (%) from cryoconite of Nepali Himalaya (Yala glacier), Greenland (Thule glacier, Isunnguata Sermia glacier, Qaanaaq glacier).

| Glacier Name | GPS (Latitude Longitude) and Elevation | Strain | Sequence Deposition no. | Total Sequence Length After Alignment | No. of Base Changes | 16S rRNA Gene Sequences Similarity (%) |
|---|---|---|---|---|---|---|
| Yala glacier, Nepal | 28°24′53″ N 85°36′48″ E Elevation: 5207 m | YNC-1 YNC-9 | MK248070 MF977331 | 1414 1405 | 5 24 | *Bacillus aryabhattai* B8W22(T) (EF114313), 98.29–99.86% |
| | 28°24′27″ N 85°36′36″ E Elevation: 5189 m | YNC-4 | MF977326 | 1491 | 3 | *Bacillus simplex* NBRC 15720(T) (NR_042136), 99.80% |
| | 28°24′27″ N 85°36′36″ E Elevation: 5189 m | YNC-34 | MK248082 | 1303 | 3 | *Brevundimonas vesicularis* NBRC 12165(T), (NR_113586), 99.77% |
| | 28°24′53″ N 85°36′48″ E Elevation: 5207 m | YNC-32 | MK248080 | 1306 | 1 | *Cryobacterium luteum* Hh 15 (HQ845193), 99.92% |
| | 28°24′27″ N 85°36′36″ E Elevation: 5189 m | YNC-3 YNC-5 YNC-7 YNC-11 YNC-25 YNC-26 YNC-27 YNC-28 | MF977325 MF977327 MF977329 MF977333 MK248075 MK248076 MF977342 MK248077 | 1345 1403 1404 1411 1198 1327 1403 1402 | 18 27 23 35 29 16 23 22 | *Cryobacterium psychrotolerans* 0549 (DQ515963), 97.58–98.66% |
| | 28°24′53″ N 85°36′48″ E Elevation:5207 m | YNC-10 YNC-18 YNC-21 YNC-22 YNC-23 YNC-29 YNC-30 YNC-35 YNC-38 | MF977332 MF977336 MF977338 MF977339 MF977340 MK248078 MF977343 MF977344 MF977345 | 1409 1409 1401 1407 1404 1401 1410 1409 1406 | 33 36 19 27 24 20 38 36 26 | *Cryobacterium psychrotolerans* 0549 (DQ515963), 97.30–98.79% |

**Table 1.** *Cont.*

| Glacier Name | GPS (Latitude Longitude) and Elevation | Strain | Sequence Deposition no. | Total Sequence Length After Alignment | No. of Base Changes | 16S rRNA Gene Sequences Similarity (%) |
|---|---|---|---|---|---|---|
| | 28°24'27'' N 85°36'36'' E Elevation: 5189 m | YNC-14 YNC-20 | MF977335 MF977337 | 1457 1456 | 9 7 | *Dermacoccus nishinomiyaensis* DSM 20448(T) (X87757), 99.38–99.52% |
| | 28°24'27'' N 85°36'36'' E Elevation: 5189 m | YNC-8 | MF977330 | 1478 | 60 | *Glaciihabitans tibetensis* MP203(T) (KC256953), 95.94% |
| | 28°24'53'' N 85°36'48'' E Elevation: 5207 m | YNC-15 YNC-24 YNC-3 YNC-33 | MK248071 MK248074 MK248079 MK248081 | 843 832 832 797 | 11 11 11 25 | *Leifsonia kafniensis* KFC-22(T) (AM889135), 96.86–98.70% |
| | 28°24'27'' N 85°36'36'' E Elevation: 5189 m | YNC-19 | MK248073 | 843 | 11 | |
| | 28°24'53'' N 85°36'48'' E Elevation: 5207 m | YNC-16 | MK248072 | 1169 | 13 | *Paracoccus limosus* (HQ336256), 98.89% |
| | 28°24'27'' N 85°36'36'' E Elevation: 5189 m | YNC-2 YNC-6 | MF977324 MF977328 | 1481 1473 | 16 15 | *Polaromonas glacialis* Cr4-12(T) (HM583568), 99.45–99.98% |
| | 28°24'53'' N 85°36'48'' E Elevation: 5207 m | YNC-36 | MK248083 | 1403 | 8 | *Sporosarcina globispora* DSM 4(T), (X68415), 99.43% |
| | 28°24'27'' N 85°36'36'' E Elevation: 5189 m | YNC-12 | MF977334 | 1500 | 6 | *Staphylococcus saprophyticus* subsp.*saprophyticus* ATCC 15305(T) (NR_074999) 99.60% |
| | 28°24'27'' N 85°36'36'' E Elevation: 5189 m | YNC-37 | MK248084 | 1394 | 26 | *Variovorax ginsengisoli* Gsoil 3165(T), (AB245358), 98.13% |

**Table 1.** *Cont.*

| Glacier Name | GPS (Latitude Longitude) and Elevation | Strain | Sequence Deposition no. | Total Sequence Length After Alignment | No. of Base Changes | 16S rRNA Gene Sequences Similarity (%) |
|---|---|---|---|---|---|---|
| Thule glacier, Greenland | 76°24′17″ N 69°43′18″ W Elevation: 536 m | TGC-2 | MF977346 | 1263 | 10 | *Cryobacterium psychrotolerans* 0549 (DQ515963), 98.36–99.21–% |
| | | TGC-5 | MK248088 | 1328 | 20 | |
| | | TGC-7 | MK248090 | 1328 | 20 | |
| | | TGC-8 | MK248091 | 1326 | 18 | |
| | | TGC-9 | MK248092 | 1300 | 22 | |
| | | TGC-10 | MF977347 | 1402 | 23 | |
| | 76°24′17″ N 69°43′18″ W Elevation: 536 m | TGC-4 | MK248087 | 1454 | 6 | *Dermacoccus nishinomiyaensis* DSM 20448(T) (X87757), 99.59% |
| | 76°24′17″ N 69°43′18″ W Elevation: 536 m | TGC-1 | MK248085 | 1309 | 7 | *Enhydrobacter aerosaccus* LMG 21877(T) (AJ550856), 99.47% |
| | 76°24′17″ N 69°43′18″ W Elevation: 536 m | TGC-3 | MK248086 | 1393 | 44 | *Glaciihabitans tibetensis* MP203(T), (KC256953), 96.70–96.84% |
| | | TGC-6 | MK248089 | 1386 | 42 | |
| Isunnguata Sermia glacier, Greenland | 76°11′05″ N 51°58′41″ W Elevation: 581 m | RGC-6 | MF977348 | 1407 | 33 | *Cryobacterium psychrotolerans* 0549(DQ515963), 97.65–99.05% |
| | | RGC-7 | MK248097 | 1402 | 22 | |
| | | RGC-8 | MK248098 | 1392 | 20 | |
| | | RGC-11 | MK246105 | 1401 | 20 | |
| | | RGC-12 | MK246106 | 1401 | 21 | |
| | | RGC-13 | MK246107 | 1401 | 21 | |
| | 76°10′35″ N 51°57′59″ W Elevation: 575 m | RGC-10 | MK246104 | 1338 | 14 | *Cryobacterium psychrotolerans* 0549 (DQ515963), 98.95% |
| | 76°11′05″ N 51°58′41″ W Elevation: 581 m | RGC-5 | MK248096 | 1353 | 5 | *Dermacoccus nishinomiyaensis* DSM 20448(T), (X87757), 99.63% |

**Table 1.** *Cont.*

| Glacier Name | GPS (Latitude Longitude) and Elevation | Strain | Sequence Deposition no. | Total Sequence Length After Alignment | No. of Base Changes | 16S rRNA Gene Sequences Similarity (%) |
|---|---|---|---|---|---|---|
| Qaanaaq glacier, Greenland | 76°10′35″ N 51°57′59″ W Elevation: 575 m | RGC-2 RGC-3 RGC-4 RGC-9 | MK248093 MK248094 MK248095 MK246103 | 1006 1345 1361 1454 | 25 26 27 37 | *Subtercola frigoramans* K265(T) (AF224723), 97.46–98.07% |
| | 77°30′12″ N 70°51′15″ W Elevation: 668 m | QGC-12 | MK246119 | 1480 | 0 | *Bacillus cereus* ATCC 14579(T), (NR_114582), 100.00% |
| | 77°30′12″ N 70°51′15″ W Elevation: 668 m | QGC-6 QGC-7 QGC-8 QGC-9 | MK246113 MK246114 MK246115 MK246116 | 1259 1401 1401 1402 | 15 20 20 23 | *Cryobacterium psychrotolerans* 0549 (DQ515963) 98.36–98.81% |
| | 77°29′27″ N 70°44′57″ W Elevation: 247 m | QGC-10 QGC-13 | MK246117 MK246120 | 1363 1401 | 15 20 | *Cryobacterium psychrotolerans* 0549(DQ515963), 98.57–98.90% |
| | 77°30′12″ N 70°51′15″ W Elevation: 668 m | QGC-15 QGC-16 | MK246122 MK246123 | 1328 1488 | 42 49 | *Glaciihabitans tibetensis* MP203(T), (KC256953), 96.71–96.84% |
| | 77°29′27″ N 70°44′57″ W Elevation: 247 m | QGC-1 QGC-2 QGC-3 QGC-4 QGC-11 QGC-14 | MK246108 MK246109 MK246110 MK246111 MK246118 MK246121 | 1393 1450 1453 1446 1454 1406 | 26 16 18 19 19 17 | *Subtercola frigoramans* K265(T), (AF224723), 98.13–98.90% |
| | 77°30′12″ N 70°51′15″ W Elevation: 668 m | QGC-5 QGC-17 | MK246112 MK246124 | 1444 1453 | 17 17 | *Subtercola frigoramans* K265(T)(AF224723), 98.82–98.83% |

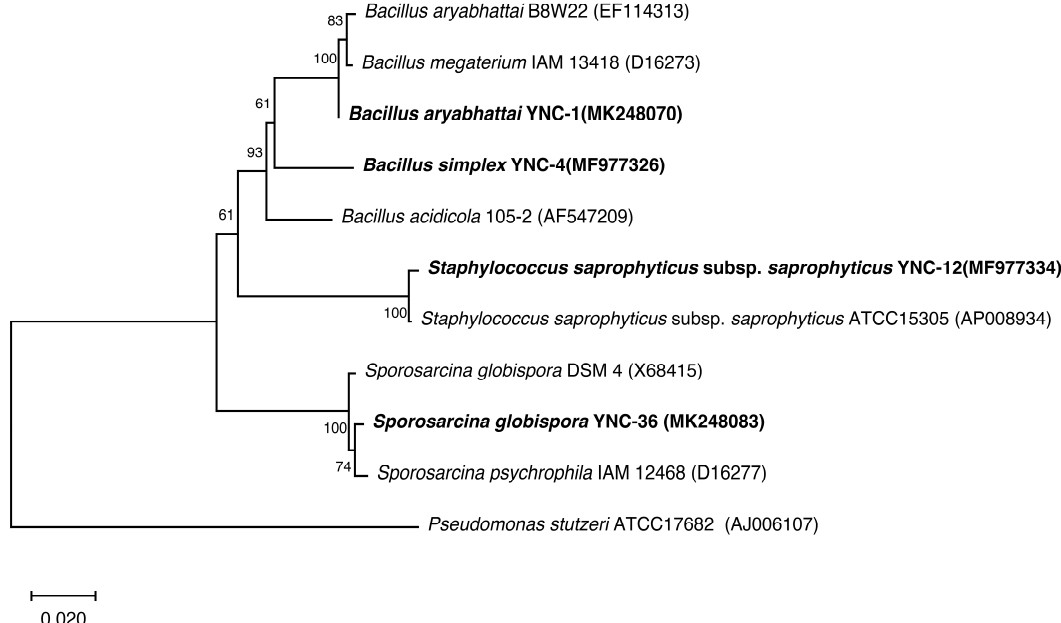

**(a)**

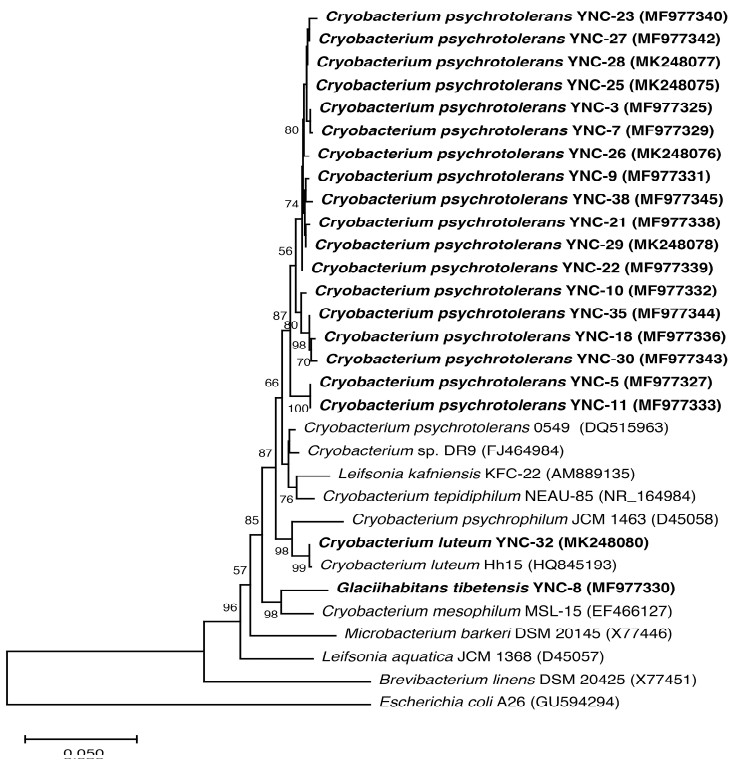

**(b)**

**Figure 2.** *Cont.*

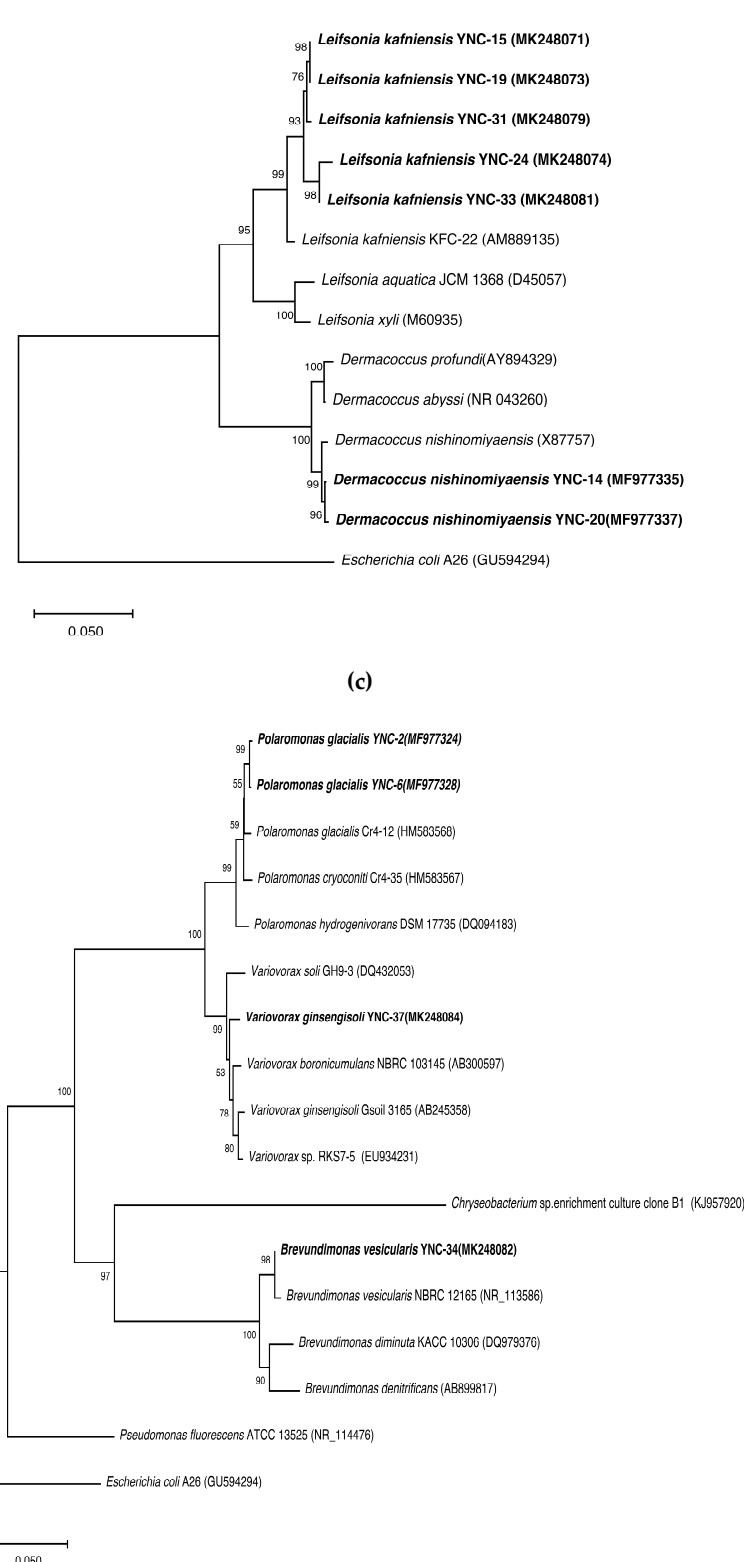

**(c)**

**(d)**

**Figure 2.** Phylogenetic tree of Bacterial strains isolated from Yala glacier, Himalaya and their closely related reference species based on the 16S rRNA gene sequences. Accession numbers are in parentheses. Tree was constructed using neighbor-joined method. (**a**) Strains of genera *Bacillus*, *Staphylococcus* and *Sporosarcina* (**b**) Strains of genera *Cryobacterium* and *Glaciihabitans,* (**c**) Strains of genera *Leifsonia* and *Dermacoccus,* (**d**) Strains of genera *Brevundimonas*, *Polaromonas* and *Variovorax.*

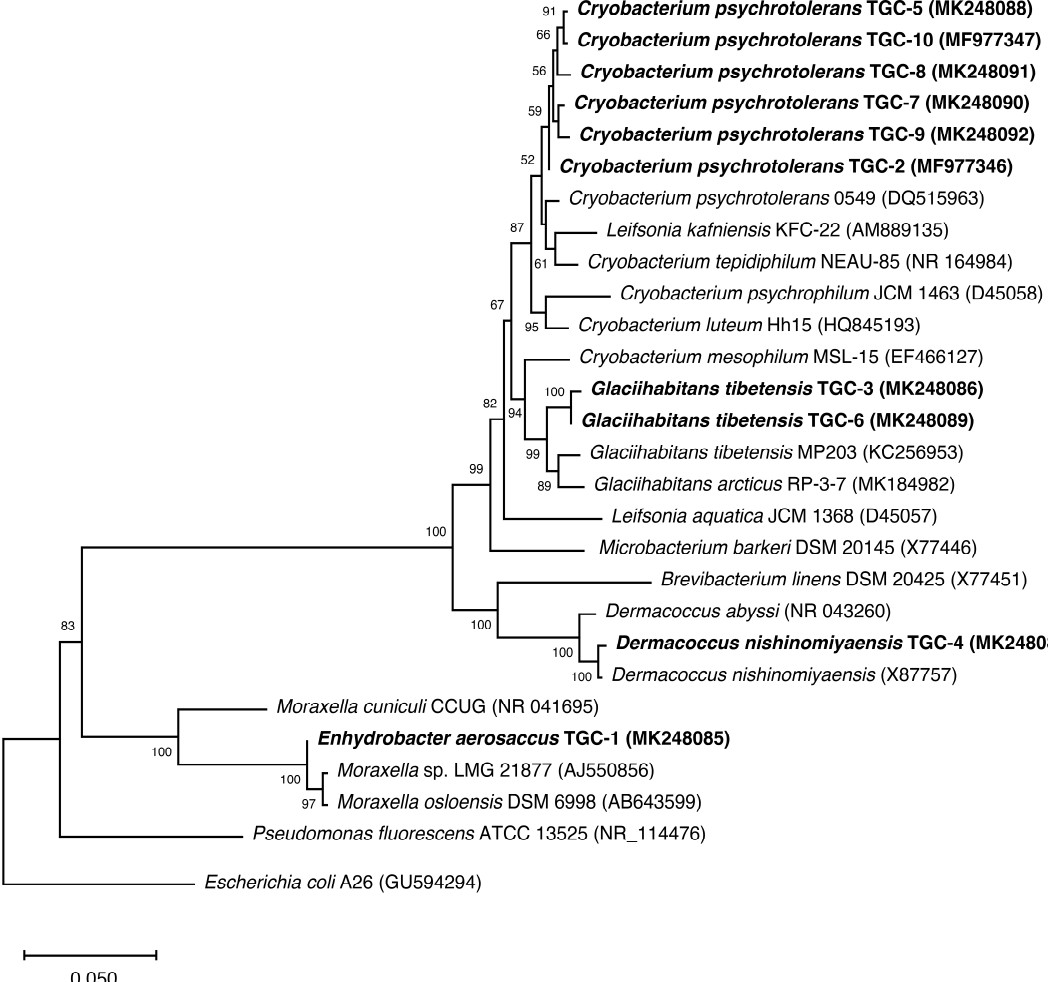

**Figure 3.** Phylogenetic tree of Bacterial strains of genera *Cryobacterium, Dermacoccus, Enhydrobacter* and *Glaciihabitans* analyzed from Thule glacier, Greenland, and reference species based on the 16S rRNA. The accession numbers are in parentheses. Tree was constructed using neighbor-joined method; each branch indicates a bootstrap value, and scale bar is estimated substitutions per nucleotide position.

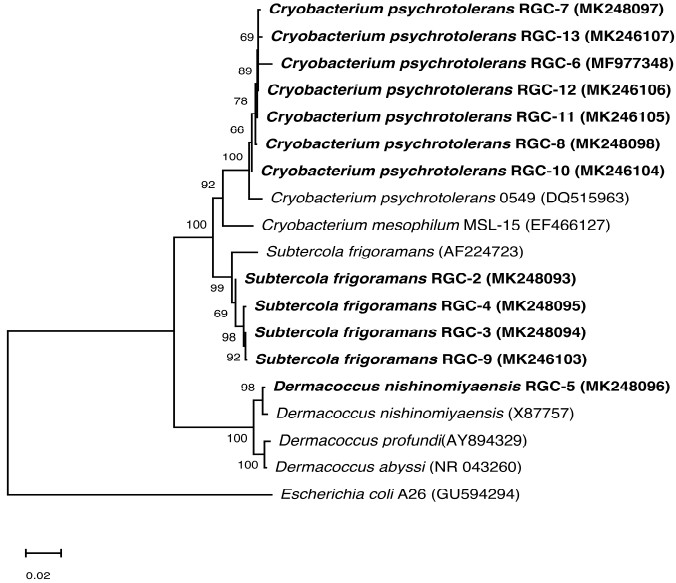

**Figure 4.** Phylogenetic tree of Bacterial strains of genera *Cryobacterium, Dermacoccus,* and *Subtercola* analyzed from Isunnguata Sermia glacier, Greenland, and reference species based on the 16S rRNA gene sequences. The accession numbers are shown in parentheses. Tree was constructed using neighbor-joined method; each branch indicates a bootstrap value, and scale bar is estimated substitutions per nucleotide position.

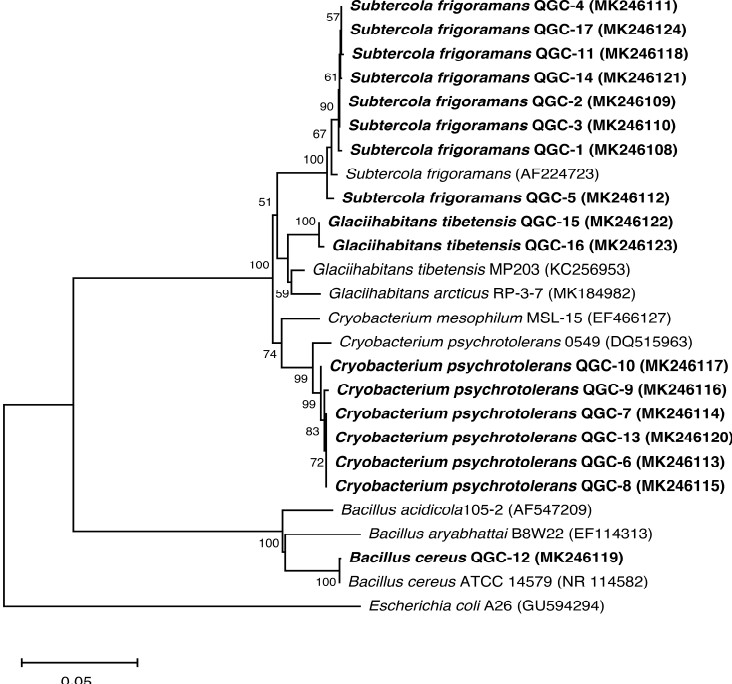

**Figure 5.** Phylogenetic tree of Bacterial strains of genera *Bacillus, Cryobacterium, Glaciihabitans,* and *Subtercola* analyzed from Qaanaaq glacier, Greenland, and reference species based on the 16S rRNA gene sequences. The accession numbers of strains are shown in parentheses. Tree was constructed using neighbor-joined method; each branch indicates a bootstrap value, and scale bar is estimated substitutions per nucleotide position.

**Table 2.** Species names, strain names, sampling point, and NCBI-GenBank accession number of the ITS region or D1/D2 domain sequences similarity (%) of isolated yeasts of Nepali Himalaya and Greenland, Arctic.

| Identity Based on ITS and D1/D2 Gene | Sampling Region | Sampling Location (Latitude Longitude) | Strain | Accession Number | Total Sequence Length after Alignment | No. of Base Changes | ITS region and D1/D2 Sequences Similarity (%) |
|---|---|---|---|---|---|---|---|
| **Filobasidiales** | | | | | | | |
| *Goffeauzyma gilvescens* | Yala glacier, Nepal Himalaya | 28°24′27″ N, 85°36′36″ E Elevation: 5189 m | J-20 | KY782275 | 531 | 0 | 100% with the D1/D2 region of *Goffeauzyma gilvescens* CBS 7525T(AF181547) |
| | Thule glacier, Greenland | 76°24′17″ N 69°43′18″ W Elevation: 536 m | J-25 | KY782276 | 530 | 0 | 100% with the D1/D2 region of *Goffeauzyma gilvescens* CBS 7525T (AF181547) |
| *Naganishia vaughanmartiniae* | Qaanaaq glacier, Greenland | 77°29′27″ N 70°44′57″ W Elevation: 247 m | J-50 | KY782277 | 364 | 1 | 99.73% with the D1/D2 region of *Naganishia vaughanmartiniae* CBS13,731 (KY108619) |
| *Piskurozyma fildesensis* | Isunnguata Sermia glacier, Greenland | 76°10′35″ N 51°57′59″ W Elevation: 575 m | J-40 J-237 | KY782278 KY782279 | 579 587 | 0 | 100% with the D1/D2 region of *Piskurozyma fildesensis* CBS12705 (KC894160) |
| **Kriegeriales** | | | | | | | |
| *Rhodotorula svalbardensis* | Isunnguata Sermia, Greenland | 76°11′05″ N 51°58′41″ W Elevation: 581 m | J-131 J-174 J-216 | KY782281 KY782282 KY782284 | 1180 1165 1169 | 11 9 72 | 93.84–99.07% with ITS region and D1/D2 domain of *Rhodotorula svalbardensis* MLB-I (JF805370) |

**Table 2.** *Cont.*

| Identity Based on ITS and D1/D2 Gene | Sampling Region | Sampling Location (Latitude Longitude) | Strain | Accession Number | Total Sequence Length after Alignment | No. of Base Changes | ITS region and D1/D2 Sequences Similarity (%) |
|---|---|---|---|---|---|---|---|
| *Rhodotorula svalbardensis* | Qaanaaq glacier, Greenland | 77°30′12″ N 70°51′15″ W Elevation: 668 m | J-181 J-112 | KY782283 KY782280 | 1175 1166 | 10 21 | 99.15–98.20% with ITS Region and D1/D2 domain of *Rhodotorula svalbardensis* MLB-I (JF805370) |
| **Cystofilobasidiales** | | | | | | | |
| *Mrakia robertii* | Yala glacier, Himalaya | 28°24′27″ N 85°36′36″ E Elevation: 5189 m | J-65 J-66 J-120 J-121 J-221 | KY782285 KY782286 KY782297 KY782298 KY782307 | 596 587 587 589 588 | 1 1 1 1 1 | 99.83% with the D1/D2 region of *Mrakia robertii* CBS8912T (AY038811) |
| *Mrakia robertii* | Yala glacier, Himalaya | 28°24′53″ N 85°36′48″ E Elevation: 5207 m | J-67 J-113 J-117 J-225 | KY782287 KY782295 KY782296 KY782308 | 587 595 594 589 | 3 1 1 1 | 99.49–99.83% with the D1/D2 region of *Mrakia robertii* CBS8912T (AY038811) |
| *Mrakia robertii* | Isunnguata Sermia glacier, Greenland | 76°10′35″ N 51°57′59″ W Elevation: 575 m | J-82 J-86 J-92 J-127 J-229 J-31 J-34 J-36 J-39 J-89 J-93 J-94 J-130 J-205 J-209 | KY782288 KY782289 KY782290 KY782299 KY782309 KY782310 KY782311 KY782312 KY782313 KY782314 KY782315 KY782316 KY782317 KY782318 KY782319 | 589 576 595 589 587 588 595 596 588 595 593 596 596 594 592 | 4 4 1 1 1 2 2 2 2 3 3 3 3 2 3 | 99.31–99.83% with the D1/D2 region of *Mrakia robertii* CBS8912T(AY038811) |

**Table 2.** *Cont.*

| Identity Based on ITS and D1/D2 Gene | Sampling Region | Sampling Location (Latitude Longitude) | Strain | Accession Number | Total Sequence Length after Alignment | No. of Base Changes | ITS region and D1/D2 Sequences Similarity (%) |
|---|---|---|---|---|---|---|---|
| *Mrakia robertii* | Qaanaaq glacier, Greenland | 77°29′27″ N 70°44′57″ W Elevation: 247 m | J-102 | KY782291 | 590 | 1 | 99.66–99.83% with the D1/D2 region of *Mrakia robertii* CBS8912T (AY038811) |
| | | | J-103 | KY782292 | 596 | 2 | |
| | | | J-104 | KY782293 | 594 | 1 | |
| | | | J-105 | KY782294 | 596 | 1 | |
| | | | J-133 | KY782300 | 596 | 1 | |
| | | | J-134 | KY782301 | 596 | 2 | |
| | | | J-135 | KY782302 | 596 | 2 | |
| | | | J-136 | KY782303 | 589 | 1 | |
| | | | J-138 | KY782304 | 596 | 1 | |
| | | | J-139 | KY782305 | 592 | 1 | |
| | | | J-214 | KY782306 | 589 | 2 | |

**Table 3.** Species names, strain names, sampling point, and NCBI-GenBank accession number of the ITS region and D1/Dr2 domain sequences similarity (%) of isolated filamentous fungi of Nepali Himalaya and Greenland, Arctic.

| Identity Based on ITS and D1/D2 Gene | Glacier Name | GPS (Latitude Longitude) and Elevation | Strain | Sequence Deposition No. | Total Sequence Length after Alignment | No. of Base Changes | Sequences Similarity (%) with Database |
|---|---|---|---|---|---|---|---|
| Dothideomycetes | Yala, Nepali Himalaya | 28°24′53″ N 85°36′48″ E Elevation: 5207 m | J-72 | MF043961 | 1067 | 2 | 99.81% with ITS region and D1/D2 domain of Dothideomycetes sp. G2-4-2 (LC514932) |
| Helotiales (*Phialophora* sp.) | Yala, Nepali Himalaya | 28°24′27″ N 85°36′36″ E Elevation: 5189 m | J-147 J-149 | MF043964 MF043965 | 1089 1096 | 18 18 | 98.17–98.36% with ITS region and D1/D2 domain of *Phialophora* sp. MLB-Phi(JN113039) |
| | | 28°24′53″ N, 85°36′48″ E Elevation: 5207 m | J-150 | MF043966 | 1092 | 20 | |
| Helotiales (*Phialophora* sp.) | Thule, Greenland | 76°24′17″ N 69°43′18″ W Elevation: 536 m | J-161 J-162 J-165 J-166 J-167 J-168 J-171 | MF043967 MF043968 MF043969 MF043970 MF043971 MF043972 MF043973 | 1095 1098 1088 1092 1097 1097 1086 | 15 19 18 18 18 19 19 | 98.25–98.63% with ITS region and D1/D2 domain of *Phialophora* sp. MLB-Phi(JN113039) |
| Leotiales (*Articulospora* sp.) | Isunnguata Sermia, Greenland | 76°10′35″ N 51°57′59″ W Elevation: 575 m | J-37 J-41 | MF043957 MF043958 | 1066 1066 | 30 30 | 98.19% with ITS region and D1/D2 domain of *Articulospora tetracladia* EF18 (LC131000) |
| Helotiales (*Phialophora* sp.) | Isunnguata Sermia, Greenland | 76°11′05″ N 51°58′41″ W Elevation: 581 m | J-175 | MF043974 | 1106 | 19 | 98.29% with ITS region and D1/D2 domain of *Phialophora* sp. MLB-Phi(JN113039) |

**Table 3.** *Cont.*

| Identity Based on ITS and D1/D2 Gene | Glacier Name | GPS (Latitude Longitude) and Elevation | Strain | Sequence Deposition No. | Total Sequence Length after Alignment | No. of Base Changes | Sequences Similarity (%) with Database |
|---|---|---|---|---|---|---|---|
| Dothideomycetes | Qaanaaq, Greenland | 77°29′27″ N 70°44′57″ W Elevation: 247 m | J-49 | MF043960 | 1072 | 0 | 99.81–100% with ITS region and D1/D2 domain of Dothideomycetes sp. G2-4-2 (LC514932) |
| | | 77°30′12″ N 70°51′15″ W Elevation: 668 m | J-140 | MF043963 | 1071 | 2 | |
| Helotiales (*Alatospora acuminata*) | Qaanaaq, Greenland | 77°30′12″ N 70°51′15″ W Elevation: 668 m | J-182 | MF043976 | 1093 | 73 | 93.32% with ITS region and D1/D2 domain of *Alatospora acuminate* DSM105,546 (MK353088) |
| Thelebolales (*Thelebolus microspores*) | Qaanaaq, Greenland | 77°29′27″ N 70°44′57″ W Elevation: 247 m | J-48 | MF043959 | 481 | 4 | 99.17% with ITS region of *Thelebolus microspores* CBS137501(AY957552) |
| | | 77°30′12″ N 70°51′15″ W Elevation: 668 m | J-245 | MF043977 | 481 | 4 | 99.17% with ITS region of *Thelebolus microspores* CBS137501(AY957552) |

The sequence analyses of the D1/D2 and ITS regions of 26S rRNA gene of fungal strains (isolates) showed five species of yeasts belonging to three orders such as Filobasidiales (*Goffeauzyma gilvescens*, *Naganishia vaughanmartiniae, Piskurozyma fildesensis*), Kriegeriales (*Rhodotorula svalbardensis*), Cystofilobasidiales (*Mrakia robertii*), and four species of filamentous fungi affiliated to orders, namely, Leotiales (*Articulospora* sp.), Thelebolales (*Thelebolus microspores*) and Helotiales (*Alatospora acuminata, Phialophora* sp.) were recorded from Himalaya and Greenland glaciers. A few strains of filamentous fungi showed similarity only at class level (Dothideomycetes) and indicated identity 99.81–100% with Dothideomycetes sp. G2-4-2 (LC514932). The affiliations of different strains of fungi with database have been elaborated. *Glaciozyma antarctica* CBS5942 was used as outgroup and a phylogenetic tree of yeast isolates was constructed (Figure 6a,b). *Curvularia lunata* JP89B-1X was used as outgroup and a phylogenetic tree of filamentous fungi was constructed (Figure 7).

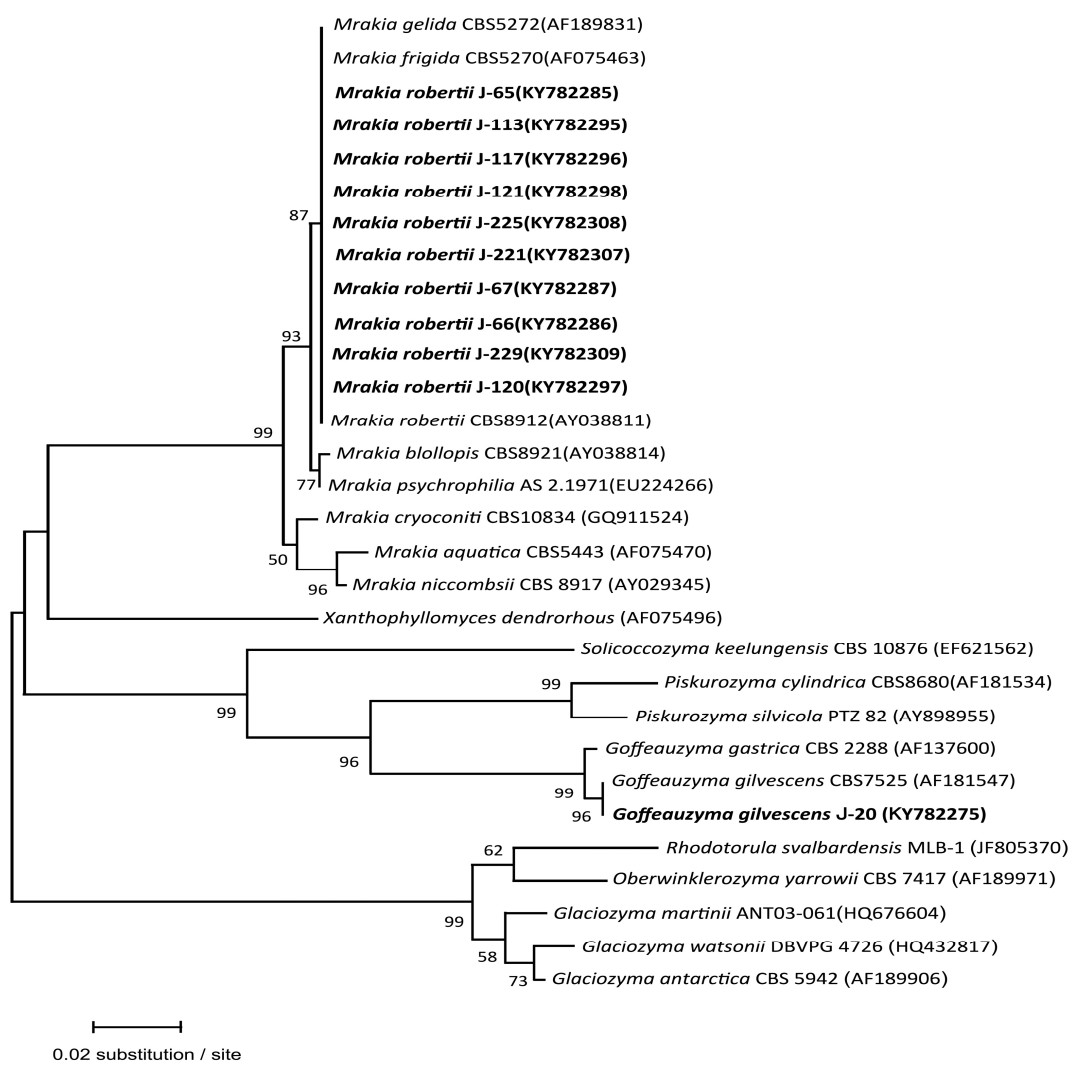

**(a)**

**Figure 6.** *Cont.*

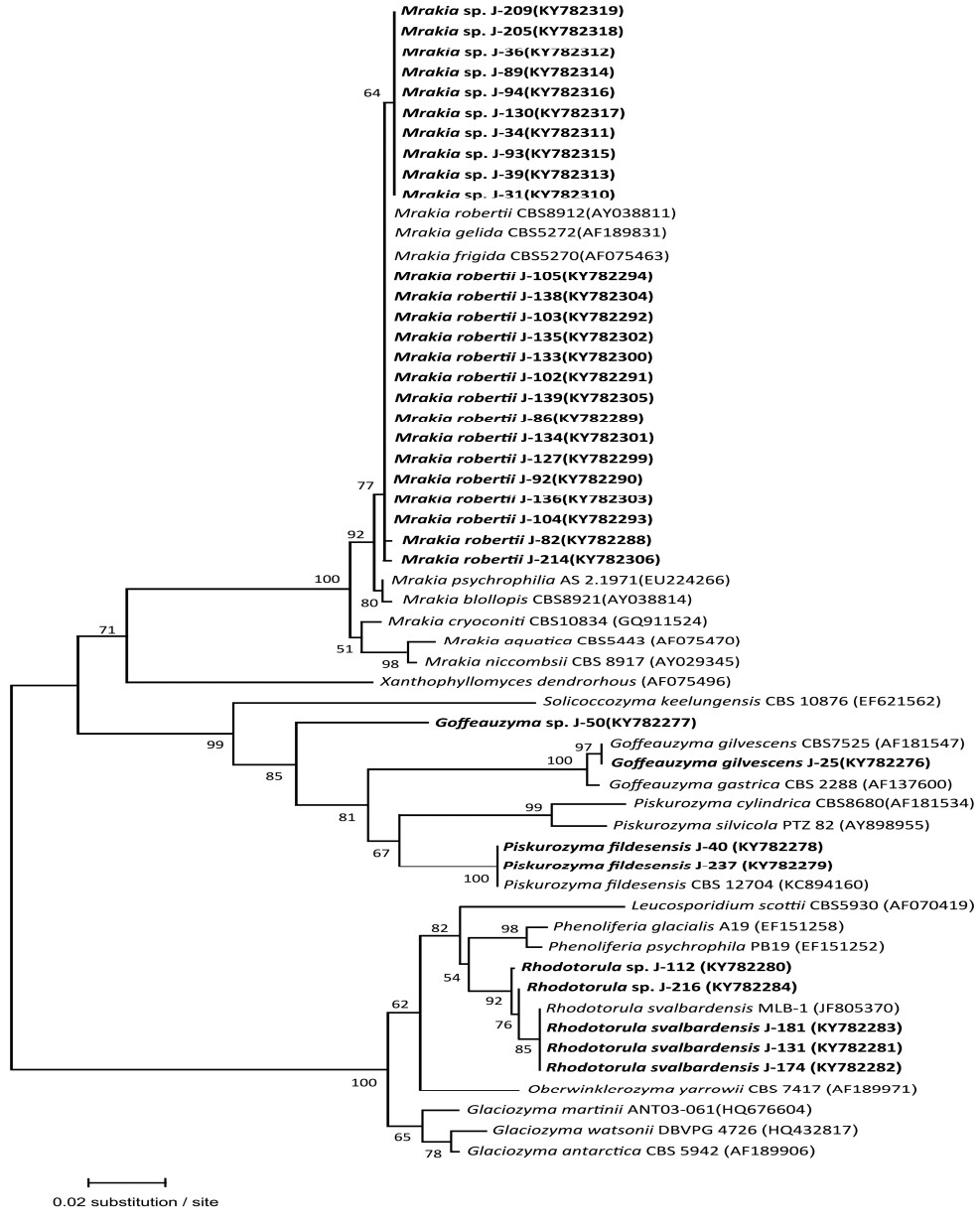

**(b)**

**Figure 6.** Phylogenetic tree of yeast species analyzed from: (**a**) Yala glacier, Himalaya, (**b**) Greenland glaciers (Qaanaaq, Thule and Isunnguata Sermia), The closely related species are based on ITS or D1/Dr2 domain of sequences. The accession numbers of strains are shown in parentheses. Tree was constructed using neighbor-joined method; each branch indicates a bootstrap value, and scale bar is estimated substitutions per nucleotide position.

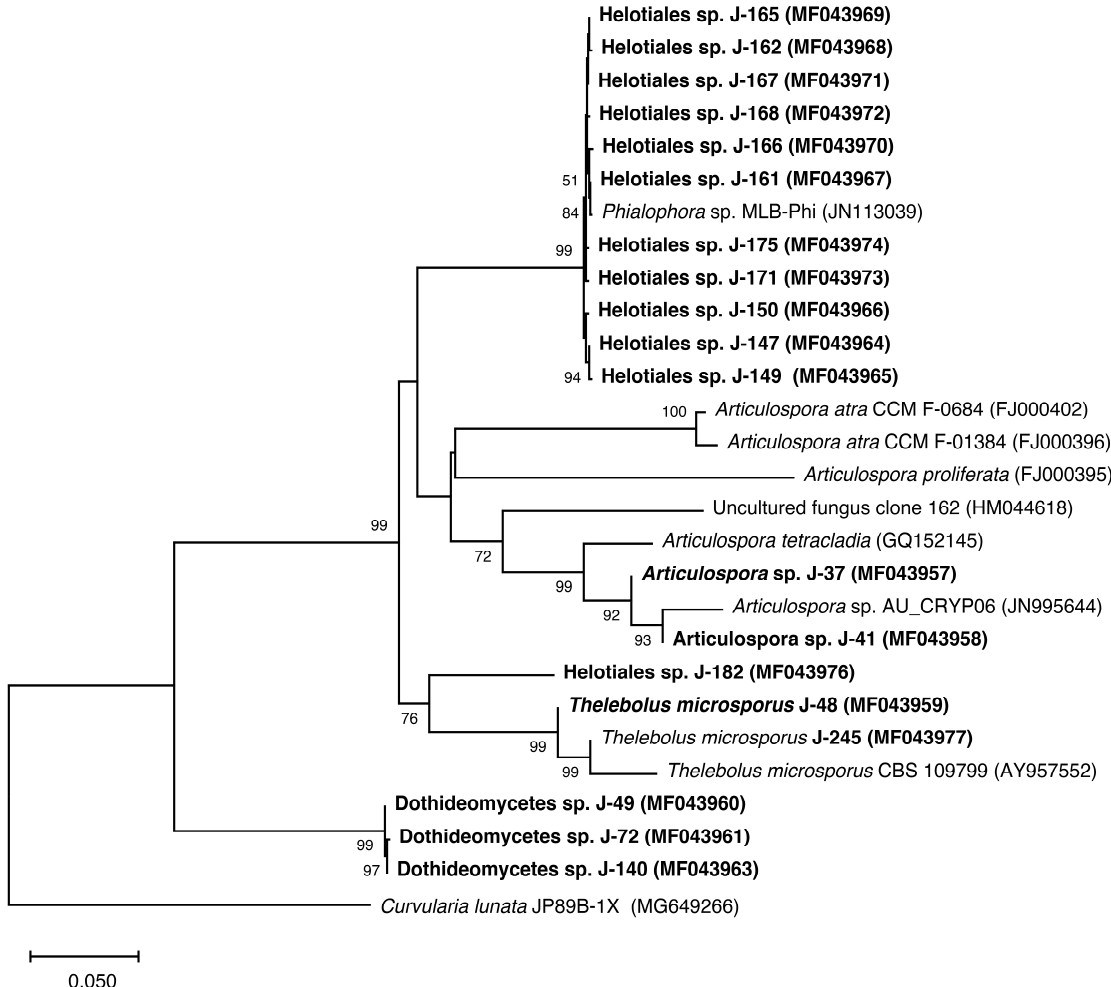

**Figure 7.** Phylogenetic tree of filamentous fungi analyzed from Nepali Himalaya and Greenland glaciers, and closely related species based on ITS region and D1/Dr2 domain of sequences. The accession numbers of strains are shown in parentheses. Tree was constructed using neighbor-joined method; significance of each branch is indicated by a bootstrap value, and scale bar is estimated substitutions per nucleotide position.

Comparative analyses of fungi from four glaciers belonging to Himalaya and Greenland showed contrasting pattern in species composition. Four species such as *Goffeauzyma gilvescens*, *Mrakia robertii*, *Phialophora* sp. (Helotiales) and Dothideomycetes sp. were recorded from Yala glacier, Himalaya (Tables 2 and 3). In contrast to this, at Greenland, two species such as *Goffeauzyma gilvescens* and *Phialophora* sp. (Helotiales) were recorded from Thule glacier. Five species such as *Piskurozyma fildesensis*, *Rhodotorula svalbardensis*, *Mrakia robertii*, *Articulospora* sp. (Leotiales) and *Phialophora* sp. (Helotiales) were analyzed from Isunnguata Sermia glacier. Five species such as *Naganishia vaughanmartiniae*, *Rhodotorula svalbardensis*, *Mrakia robertii*, *Thelebolus microspores*, *Alatospora acuminate* (Helotiales) and strains resembled to class Dothideomycetes were recorded from Qaanaaq glacier (Tables 2 and 3).

### 3.4. Distribution Patterns of the Bacteria and Fungi in Four Glaciers' Cryoconite of Greenland and Himalaya

The bacterial species isolated during current study from Himalaya and Greenland cryoconite granules belonged to 13 genera such as *Bacillus*, *Brevundimonas*, *Cryobacterium*, *Dermacoccus*, *Enhydrobacter*, *Glaciihabitans*, *Leifsonia*, *Paracoccus*, *Polaromonas*, *Sporosarcina*, *Staphylococcus*, *Subtercola* and *Variovorax* (Table 1). Number of bacterial genera at each glacier varied: 11 from Yala glacier,

4 from Thule glacier, 3 from Isunnguata Sermia glacier and 4 from Qaanaaq glacier. *Cryobacterium psychrotolerans* was common to all glaciers' cryoconite. *Subtercola frigoramans*, *Glaciihabitans tibetensis* and *Dermacoccus nishinomiyaensis* were the next most abundant species. *Brevundimonas vesicularis*, *Leifsonia kafniensis*, *Paracoccus limosus*, *Polaromonas glacialis*, *Sporosarcina globispora*, *Staphylococcus saprophyticus* and *Variovorax ginsengisoli* were only present in the Himalaya and not on the GrIS (TG, IS and QG). *Enhydrobacter aerosaccus* was present in TG and absent in YG, IS, QG glaciers. Genera *Brevundimonas*, *Enhydrobacter*, *Leifsonia*, *Paracoccus*, *Sporosarcina*, *Staphylococcus* and *Variovorax* were represented the least. *Bacillus aryabhattai*, *Bacillus simplex* and *Cryobacterium luteum* were present only in YG glacier; *Bacillus cereus* was present only in QG glacier. *Glaciihabitans tibetensis* was represented in YG, TG and QG glaciers and *Dermacoccus nishinomiyaensis* was present in YG, TG, and IS glaciers. Glacier YG indicated maximum bacterial diversity representing 13 species while IS glacier showed least species diversity, having only 3 species. The current observations reveal that distribution of bacterial species exhibit contrasting patterns along Greenland and Himalayan glaciers' cryoconite.

The yeasts and filamentous fungi from the cryoconite granules of Greenland and Nepali Himalayan glaciers belonged to nine genera namely, *Goffeauzyma, Mrakia, Naganishia, Piskurozyma, Rhodotorula, Alatospora, Articulospora, Phialophora, Thelebolus* (Tables 2 and 3). Besides these, a few isolates of many filamentous fungi also belong to class Dothideomycetes. Number of fungal genera at each glacier varied: 3 from Yala glacier, 2 from Thule glacier, 5 from Isunnguata Sermia glacier and 5 from Qaanaaq glacier. The most dominant species was *Mrakia robertii* followed by strains of *Phialophora* sp. (Helotiales). *Goffeauzyma gilvescens* were distributed at YG and TG glaciers. *Piskurozyma fildesensis* was present only in IS glacier and absent in YG, TG and QG glaciers. *Naganishia vaughanmartiniae* was represented only at QG and absent in YG, TG and IS glaciers. *Rhodotorula svalbardensis* was present in IS and QG glaciers, and absent at YG and TG. Genera *Alatospora, Articulospora* and *Thelebolus* were represented the least. *Articulospora* was represented in IS, and absent at QG, YG and TG glaciers. Similarly, *Alatospora* represented only at QG. Dothideomycetes were presented in YG and QG glaciers. Glacier IS and QG showed maximum diversity representing five species each, while TG glacier indicated least diversity having only two species.

## 4. Discussion

Amplicon sequence-based approach in diversity analyses documented a larger assortment of microbes [16,19,43–48,50,51,56], and reported bacterial species affiliated to four classes such as Proteobacteria, Cyanobacteria, Bacteroidetes, Actinobacteria, and a few fungi (Microbotryomycetes and Chytridiomycota). Culturable approach, as used in current study, has importance in physiological, biochemical, and biotechnological characterizations of individual species [34]. Further, it also helps in understanding inter- and intra-species interactions in an ecosystem [75,76]. In order to maximize the recovery of culturable microbes, numerous bacteriological and fungal medias of different strengths, and low temperature gradients, were applied in the current study. It was observed that out of 139 isolates, 48 were recovered from diluted media (1/10), indicating their oligotrophic characteristic, like Svalbard [34,35].

The bacterial and fungal isolates were able to grow at low temperature between 1 and 15 °C, thus confirming their psychrophilic nature. Similar studies have also been carried out from Antarctic [57], and Arctic cryoconite [34,35] observed that the microbes grew between 4 and 22 °C. In contrast to this, bacterial isolates from the alpine glaciers' cryoconite were psychrotolerant and grew profusely at 30 °C [58].

The bacterial species from the cryoconite of Greenland and Nepali Himalaya belonged to 13 genera namely, *Bacillus, Brevundimonas, Cryobacterium, Enhydrobacter, Glaciihabitans, Dermacoccus, Leifsonia, Polaromonas, Paracoccus, Staphylococcus, Sporosarcina, Subtercola* and *Variovorax*. Genus *Cryobacterium* has also been reported from the Antarctic cryoconite granules [57]. Genera such as *Cryobacterium, Leifsonia, Polaromonas, Subtercola* were recorded from Svalbard, Arctic [34]. Two bacterial genera (*Bacillus, Cryobacterium*) were analyzed from cryoconite of Greenland Ice Sheet [56]. Genus Bacillus

has been previously reported from cryoconite of Alpine region [58]. Further, *Polaromonas* and *Subtercola* have also been reported from extreme habitats viz. snow, permafrost and sea ice [77–80]. The yeasts (*Goffeauzyma*, *Mrakia*, *Naganishia*, *Piskurozyma*, *Rhodotorula*) and filamentous fungi (*Alatospora*, *Articulospora*, *Phialophora*, *Thelebolus*) from the cryoconite of Greenland and Nepali Himalayan glaciers belonged to nine genera. Among these, genera such as *Articulospora*, *Mrakia* and *Rhodotorula* were recorded from Svalbard, Arctic [35]. Recently, *Articulospora*, *Mrakia* and *Rhodotorula* have also been reported from the cryoconite of Greenland [56]. The occurrence of similar species at geographically distant places suggests their wide distribution, and adaptation strategies at low temperatures [34,76].

The comparison of microbial communities of four glaciers indicate that YG has a higher number of species diversity (16 species: 13 species of bacteria and 3 species of fungi) followed by QG (9 species: 4 species of Bacteria and 5 species of fungi), IS (8 species: 3 species of bacteria and 5 species of fungi) and then TG (6 species: 4 species of bacteria and 2 species of fungi). The glacier surface conditions such as elevation (Tables 1–3), mineral dust [81], and microclimate [34,35] probably limit the colonization of microbial communities despite close proximity of GrIS glaciers. Further, Cameron et al. [45] stated that bacterial communities (amplicon sequence-based) across the GrIS are spatially variable due to influence of localized biological inputs and physicochemical conditions. Himalaya has more intense solar radiation due to higher sun elevation and greater impacts of anthropogenic pollutions and terrestrial dust than Greenland [81]. The results, as presented, suggest that the Himalayan site has a higher microbial diversity than the Greenland sites.

## 5. Conclusions

Cryoconite holes are unique habitat on the cryosphere where cold-adapted microbes sustain. The current study was focused on bacterial communities of Himalayan and Greenland cryoconite through the culture-based approach. A detailed gene sequence analysis (16S rRNA, D1/D2, ITS) contributed 12 species of bacteria (*Bacillus aryabhattai, Bacillus simplex, Brevundimonas vesicularis, Cryobacterium psychrotolerans, Cryobacterium luteum, Dermacoccus nishinomiyaensis, Glaciihabitans tibetensis, Staphylococcus saprophyticus, Leifsonia kafniensis, Paracoccus limosus, Sporosarcina globispora, Variovorax ginsengisoli*), and 4 species of fungi (*Goffeauzyma gilvescens, Mrakia robertii, Dothideomycetes* sp. *Helotiales* sp.) as a new record from Himalaya. Furthermore, five species of bacteria (*Bacillus cereus, Enhydrobacter aerosaccus, Dermacoccus nishinomiyaensis, Glaciihabitans tibetensis, Subtercola frigoramans*), and seven species of fungi (*Goffeauzyma gilvescens, Mrakia robertii, Naganishia vaughanmartiniae, Piskurozyma fildesensis, Thelebolus microspores, Alatospora acuminata, Phialophora* sp. and *Dothideomycetes* sp. are as a new contribution to Greenland. Comparison of present study with Perini et al.'s [56] GrIS results showed resemblance with five genera (*Bacillus, Articulospora, Cryobacterium, Mrakia, Rhodotorula*) but indicated similarity at the species level only with two taxa (*Cryobacterium psychrotolerans, Rhodotorula svalbardensis*). Future studies on cryoconite microbes need to be focused on ecological functioning at the molecular level. Screening and biotechnological characterization of these culturable microbes may help in health, agriculture and industry.

**Supplementary Materials:** The following are available online at http://www.mdpi.com/2071-1050/12/16/6477/s1, Table S1: Bacterial culture details of Nepali Himalaya and Greenland glaciers, Table S2: Yeast culture details of Nepali Himalaya and Greenland glaciers, Table S3: Filamentous fungi culture details of Nepali Himalaya and Greenland glaciers.

**Author Contributions:** P.S. analyzes the results and writing of the original draft preparation. M.T. constructed trees. S.M.S. involved in planning of the study. N.T. contributed in sampling. All authors have read and agreed to the published version of the manuscript.

**Funding:** This research received funding from SERB (PDF/2016/003707), Department of Science and Technology (SR/WOS-A/LS-419/2013(G), NCAOR, JSPS KAKENHI (19H01143 and 20K21840), and from the Arctic Challenge for Sustainability II (ArCS II), Program Grant Number JPMXD1420318865.

**Acknowledgments:** N.T. is thankful to Department of Hydrology and Meteorology, Govt. of Nepal, P.S. thankful to SERB and DST. S.M.S. is grateful to NCAOR and BHU for support. S.M.S. is also thankful to Simantini Naik

for technical help. Authors are grateful to the reviewers for fruitful suggestions. Special thanks to Almighty for fueling us to finalize and publish this article during difficult time of pandemic Covid-19.

**Conflicts of Interest:** The authors declare no conflict of interest.

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
