# Peer review of "Contrasting Patterns of Microbial Communities in Glacier Cryoconite of Nepali Himalaya and Greenland, Arctic"

_sustainability, doi:10.3390/su12166477_

Round 1

Reviewer 1 Report

The work is devoted to the identification of bacteria and fungi contained in cryoconites. Samples were taken from three places in Greenland and in one place in Nepal, in the glaciers of the Himalayas.

This work is important for a common understanding of the spread of microorganisms and their role in ecosystems. Since there are not many researchers involved in this topic, the number of people interested in the article will not be so high, but one should not conclude that it is of poor quality - there are simply not many specialists who deal with this topic. But the topic is important and relevant.

The introduction discusses previous publications on this topic, which, in principle, are few. The manuscript authors investigated cultivated species grown on different media. Species diagnostics was carried out by analysis of ribosomal RNA genes. The results are placed in the EzBioCloud dataset (Methods section).

The Discussion section compares the species composition with previously published data. In conclusion, the authors note that it is important to focus on molecular research, as well as on screening the biotechnological properties of isolated microorganisms.

The disadvantages of the following:

In figure 1 there is no picture f).

In tables 1 a-d, the headings indicate the locations from which the samples were taken. Each table is dedicated to one such place, so it was unnecessary to report this on every row, cluttering the table.

A similar remark applies to table 2.

Please specify where the sequences were deposited. Table 3 shows the GenBank database. Other tables do not indicate. In methods line 109, only GenBank is indicated. Please specify and indicate for all tables.

Line 166 - Number 1. Why is it so marked? Where 2, 3, etc. This is mistake?

 Please arrange the captions for figure 2 a-d. Lines 332-335. The same text on lines 341-344. It is advisable to give signatures for each phylogenetic tree.

Figure 2a says: Staphylococcus saprophyticus subsp. saprophyticus ... subsp. should not be in italics.

Author Response

Reply to Reviewer Comments

Reviewer 2 Report

The authors present the identities of a number of isolates from four glaciers, three on the Greenland Ice Sheet (GrIS) and one in the Nepali Himalayas. There is a lack of such information in the publish literature, especially for the GrIS, and therefore there is some merit to this study being published.

 However, the manuscript, in its current form, does not provide sufficient information for the results to be of as much use to other scientists as they could be if better presented. I have highlighted how this could be remedied below, and after that I have listed some minor points that also need addressing.

In addition, although the English language of the manuscript is perfectly understandable, it would benefit greatly by some language editing. There are for instance several sentences without a verb, or with incorrect grammar.

MAJOR COMMENTS:

Introduction: Although the authors do a good job of introducing these supraglacial ecosystems and highlighting the diversity of cryoconite-associated microorganisms, they do not clearly state why it is necessary for us to know which culturable organisms there are in these environments. Since the majority of these microorganisms are considered unculturable by many, it is important to clarify why we should focus our attention on a small subset of the total community.

Material & Methods: It is not clear from the figures and no mention in the text of how many sites were sampled per glacier; from Figure 1, it seems to be 2 sites on Qaanaaq Glacier, 1 site on Thule Glacier, 2 OR 3 sites on Isunnguata Sermia Glacier. I cannot tell from Fig 1d how many sites were on Yala Glacier. In addition, how many cryoconite holes were sampled at each glacier site? What was the total number of cryoconite samples collected and what was the size of the collected samples?

Table 1c and Table 2: Here the authors refer to Russell Glacier, not Isunnguata Sermia Glacier as introduced in the Material and Methods. Pick one name for this subset of samples and use that consistently throughout the manuscript.

Tables: i) It is not clear to me why the bacterial isolates are grouped after glacier (Tables 1a-d), whereas for the fungi (Tables 2 and 3) isolates from all glaciers are lumped together and grouped after order. ii) Also, the order of the columns differ between Tables 2 and 3, making it more difficult to compare between tables. iii) when there is a large number of isolates from the same glacier identified as the same species and matching the same strain in the database, as is the case for Mrakia robertii in Table 2, it would make more sense to condense these together into one entry.

Rows 153 & 157: There is a large difference in the number of isolates selected for identification from each glacier, ranging from 8 to 22. This will have an impact on the diversity found and thus should be considered in the interpretation of the results.

Rows 166-203 & 221-238 & 251-260: These sections contain long lists of isolates and their assigned identities. These sorts of results are much better suited to being presented in a table format and, since the authors already have done that (see tables), this repetition is superfluous in the text. Of course the authors may wish to give a more generalised overview of the taxonomical groupings in here, as would indeed be appropriate. In doing so, they would free up a lot of space to address my next point.

The authors have not presented an overview, or any details, on the outcome of the different growth media used to obtain the different isolates. This lack of information makes it harder to evaluate the validity of the results, but also, denies others interested in isolating microorganisms from cryoconite the opportunity to learn from this study. At a minimum, this information should be included for each isolate in the tables. This is especially important, since the authors champion the importance of culture-based methods and highlight the future need for isolating microbes from the environment (i.e. row 316-322). In such circumstances, it is odd that they are not evaluating the efficiency of the various growth media used.

The authors have not presented the outcome of the different incubation temperatures either.  From row 159, it seems that ALL bacteria AND fungi from ALL glaciers had an optimum growth temperature of 15*C, which seems surprisingly uniform. I would like to the growth rate/success vs incubation temperature expanded up on in the Results section.

Row 205 onwards, rows 269-277 & row 388: Most of the species found on GrIS glaciers were also found on the Himalayan glacier, therefore there is not so much of a contrast, as a selective result due to more constraining environmental conditions on the GrIS. There is no mention of nutrient conditions, water availability, in situ temperature ranges or any other environmental conditions at the sampling sites, apart from row 388, where the authors state that “Himalaya has warmer climate and richer in nutrients than Greenland, …”, without backing it up with any data or references. This type of information should be included in the paper, either as measured by the authors themselves, or by pulling in relevant data from the literature.

Phylogenetic trees: It would be more informative to see all bacterial isolates presented in one phylogenetic tree, all fungal isolates in one and so on. However, as a minimum, in trees with isolates from more than one glacier (e.g. figures 6b and 7), the source glacier should be noted on the tree itself. It is currently impossible to see which glacier each isolate came from without cross-referencing the tables. Why is some of the trees split after glacier (fig 3,4,5), when the others were not? Again, I would prefer these in one tree, but with the glaciers highlighted, e.g. by use of boxes or markers.

Rows 282-283 & rows 295-297: Since the sampling effort was not the same across glaciers or regions and since the total number of CFUs for each site is not presented, it is very difficult to determine whether the different number of taxa is a result of an actual difference in species richness, or simply a result of the difference in sampling effort. There is also no statistical support to suggest otherwise.

Row 288: Since no statistical tests are presented, you cannot say that the distributions are significantly different. Instead you have a number of taxa that are only present in the Himalayas and not on the GrIS.

Rows 371-388: Many of the species found are truly global species found in many other habitats, such as soils. I would like to see a discussion on why the authors think that they only find these ubiquitous bacterial species on a single glacier. Is it perhaps down to the limited sampling effort, or are there some environmental or other differences between the three GrIS glaciers that mean that they have vastly different microbial community compositions despite their close proximity.

Row 312 & 315: Most of these non-culture based studies are not metagenomics studies but rather amplicon sequence-based. Thus equating non-culture based studies with metagenomics studies is incorrect.

Row 315 onwards:  What are the benefits of a culture-based approach? Also, the authors could elaborate on how the culture-based approach is different to metagenomics “fishing expeditions” for antimicrobial resistance and biotechnological potential.

Row 386: I do not understand what the authors mean when they say that that “it is now evident that microbial sustainability at Himalayan cryoconites is more than Greenland cryoconites.”. Sustainable in what way? If this is only in terms of number of taxa found among the isolates, then, again, the difference in sampling effort must be taken into account before you can state this.

MINOR COMMENTS:

Rows 60-66: Perini et al’s results mirror those of other groups and of non-culture based studies. I would like the authors of the current study to comment on the similarities and dissimilarities between their results and Perini et al’s.

Row 81: When, specifically, were samples collected from each site? Where samples collected more than once from each glacier?

Row 82 onwards: Generally speaking, cryoconite is a uncountable noun (“cryoconite was collected…”) and so it would be better to use the phrase “cryoconite granules” instead of “cryoconites”. E.g. “The cryoconite was loose, rounded and brownish-black in colour. Cryoconite granules were collected from the bottom of the cryoconite holes…” or “The cryoconite was loose, rounded and brownish-black in colour and was collected from the bottom of the cryoconite holes…”.

Row 84: How were the pipettes pre-cleaned? What was the volume of the tubes?

Row 93: What was the concentration of the saline solution? These are very low salinity environments and therefore the concentration used could severely affect what species were recovered.

Row 96: Was the cryoconite plated straight on to these RBA, PDA and MEA or was suspended in saline solution prior to plating out?

Table 1c RGC-8: data missing.

Row 153: RG is not mentioned in M&M. Change to IS or use Russell glacier in the M&M.

Row 166-172: Spurious numbered list.

Figures: i) Figure 1 is followed by Figure 6a and b,  and 7 and then Figures 2, 3 and so on. Ii) Figure 2’s legend duplicated on row 332 (the legend on row 341 onwards is in the correct place).

Row 280-297, and elsewhere: Species names and genus names should be italicised throughout.

Row 314 “a few fungi”: It would be helpful to specify which fungal species these were.

Row 336 and row 363: Spurious bullet points.

Author Response

Reply to Reviewer 2 comments

Round 2

Reviewer 2 Report

The authors have successfully addressed most of my concerns with the previous version of this manuscript, however some points still need clarification:

Rows 66-68: “The studies on culturable microbes of supraglacial ecosystems have immense importance in physiological, biochemical and biotechnological characterizations of individual species [34].” --> Instead of a generalised and vague claim of culture-based studies being of 'immense importance', the authors need to give specific examples of instances where isolates are useful in a way that genetic information is not.

Rows 82-86: The sampling and the environmental conditions at the time of sampling need to be described in more detail. Sampling dates, number of samples collected at each site, temperature at the time of sampling, and any other metadata collected such as pH and EC of the water, nutrient content of the water and so on.

Row 88: If the pipettes were sterile and not further cleaned or pre-contaminated or similar in the field, then it is sufficient to say 'Cryoconite granules were collected from bottom of cryoconite-holes by aspirating with a sterile disposable plastic pipette into sterile tubes, and stored at -20ºC until analyses.'

Tables: The tables are not rendered properly in the current PDF version, so I am not able to fully comment on these. However, it seems like the authors have succeeded in making the tables more consistently laid out.

Rows 170-172: "Large difference in the number of isolates recorded from Himalayan and Greenland cryoconites indicate the presence of microbial diversity in supraglacial habitat[D21]"  --> This change does not address my previous comment. The number of different isolates obtained from each glacier will be dependent on the number of samples collected from that glacier, as the higher the sampling effort, the more likely you are of collecting more species. Therefore, without backing this up with either data or statistical testing, it is not clear whether the difference in number of isolates is the result of an actual difference in number of species present or an artifact of the sampling strategy.

Row 386: The word sustainability is not suitable in this sentence and the sentence needs to be rephrased (regardless of the title of the journal!). The results, as presented, suggest that the Himalayan site has a higher microbial diversity than the Greenland sites.

Author Response

Please find attached file for responses for the Reviewer-2 comments.

Thank you

This manuscript is a resubmission of an earlier submission. The following is a list of the peer review reports and author responses from that submission.